# CAPTURING THE CHANNEL DEPENDENCY COMPLETELY VIA KNOWLEDGE-EPISODIC MEMORY FOR TIME SERIES FORECASTING

## ABSTRACT

The forecasting of **M**ultivariate **T**ime **S**eries (MTS) has long been an important but challenging task, and recent advancements in MTS forecasting methods try to discover both temporal and channel-wise dependencies. However, we explore the nature of MTS and observe two kinds of existed channel dependencies that current methods have difficulty capturing completely. One is the evident channel dependency, which can be captured by mixing the channel information directly, and another is the latent channel dependency, which should be captured by finding the intrinsic variable that caused the same changes within MTS. To address this issue, we introduce the knowledge and episodic memory modules, which gain the specific knowledge and hard pattern memories with a well-designed recall method, to capture the latent and evident channel dependency respectively. Further, based on the proposed memory modules, we develop a pattern memory network, which recalls both memories for capturing different channel dependencies completely, for MTS forecasting. Extensive experiments on eight datasets all verify the effectiveness of the proposed memory-based forecasting method.

## 1 INTRODUCTION

Multivariate time series (MTS) is playing an important role in a wide variety of domains, including internet services (Dai et al., 2021) , industrial devices (Finn et al., 2016; Oh et al., 2015) , health care (Choi et al., 2016b;a), , finance (Maeda et al., 2019; Gu et al., 2020) , and so on. Forecasting MTS has consistently posed a formidable challenge due to the presence of intricate temporal dependencies and diverse channel-wise dependencies. To model the temporal-dependency of MTS, many dynamic methods based on recurrent neural networks (RNNs) have been developed (Malhotra et al., 2016; Zhang et al., 2019; Bai et al., 2019; Tang et al., 2020; Yao et al., 2018). With the development of Transformer (Vaswani et al., 2017) and due to its ability to capture long-range dependencies (Wen et al., 2022; Dosovitskiy et al., 2021; Wu et al., 2021) , which is especially attractive for time series forecasting, there is a recent trend to construct Transformer based MTS forecasting methods and have achieved promising results (Li et al., 2019; Zhou et al., 2021; Wu et al., 2021; Zhou et al., 2022) in learning expressive representations for MTS forecasting tasks. Moreover, recently, the Linear model (Zeng et al., 2023) is proved to be more effective in capturing the long-range temporal dependencies within MTS for forecasting. However, despite the advanced architectural design, it is still difficult for both the Transformer based and Linear based methods to predict real world time series due to the ignorance of the correlation between different channels (Zhang & Yan, 2023) (Wu et al., 2020) (Cao et al., 2020). To fill the gap, prior research has tried to used the GNN or CNN modules to mix the inter-channel information(Wu et al., 2021) (Wu et al., 2020) for boosting the prediction performance. The recently proposed crossformer (Zhang & Yan, 2023), equipped with a Dimension-Segment-Wise (DSW) embedding and Two-Stage-Attention (TSA) layer, is able to capture the channel dependency efficiently. Unfortunately, proposed methods are all designed exclusively for improving the Transformer and RNN based models, which can not be served to more effective Linear model. In addition, most of previous methods can only focus on the evident channel dependency in our opinion, they usually fail to capture the various channel dependencies completely, which arises from the presence of both evident dependencies between different channels and latent dependencies rooted in intrinsic patterns. Since these traditional approaches struggle when it comes to capturing the intrinsic sharing patterns illustrated in Figure 1. Recognizing the importance of these

intrinsic patterns in explaining why different channels exhibit similar trends—indicating that changes in different channels may be attributed to a common factor referred to as the intrinsic pattern. Thus, we introduce a knowledge memory module to synthesize and summarize the shared intrinsic patterns hidden behind the MTS to capture the latent channel dependency. In addition to the knowledge memory module, we also propose an episodic memory module to capture the evident dependency. This module is capable of storing distinctive patterns from different channels. By incorporating these two memory modules, we can capture channel dependency from distinct perspectives, enhancing the effectiveness of our approach.

Moving beyond the constraints of previous work, we combine the knowledge memory and episodic memory module to develop a **S**tudent-like **P**attern **M**emory **Net**work (SPM-Net) as illustrated in Fig. 1. Specifically, the knowledge module is adept at extracting normal temporal pattern representations from the MTS, akin to a diligent student summarizing essential concepts from their coursework. The episodic memory module is engineered to retain hard example temporal patterns, resembling a student's practice of reviewing difficult problem in preparation for an upcoming examination. When the new MTS comes, it looks up the memory and give the top-k related patterns of different channels. Given that the top-k patterns across different channels often exhibit overlaps, this enables us to discern relationships between these channels by amalgamating the shared patterns, as exemplified in Figure 1 (further elaborated in Section 3). Thus, the channel dependency can be captured efficiently in this way. The main contributions of our work are summarized as follows:

- For MTS forecasting, we propose a student-pattern memory module consisting of a knowledge memory module and an episodic memory module, which is able to consider the channel dependencies completely within the MTS via a top-k recall method.

- We develop SPM-Net, a memory based Linear model equipped with student-pattern memory module, which can consider the channel dependency issue of MTS, thus enhancing the representation ability of Linear model.

- Experiments on eight real-world datasets illustrate the efficiency of our model on MTS forecasting task. Specifically, SPM-Net ranks top-1 among the eight models for comparison on 52 out of the 64 settings and ranks top-2 on all settings.

## 2 RELATED WORK

### 2.1 MULTIVARIATE TIME SERIES FORECASTING

In recent decades, the field of MTS forecasting has evolved significantly. It has transitioned from conventional statistical approaches such as ARIMA (Ariyo et al., 2014) and machine learning techniques like GBRT (Friedman, 2001) towards more advanced deep learning-based solutions, including Recurrent Neural Networks (Lai et al., 2018) and Temporal Convolutional Networks (Bai et al., 2018), (Liu et al., 2021a). These traditional models usually have difficulty modeling long-term dependency (Zhang & Yan, 2022). Recently, a number of Transformer-based models have been proposed for MTS forecasting and show their great ability to capture the long-range temporal dependency. For example, LogTrans (Li et al., 2019) incorporates causal convolutions into self-attention layer to consider local temporal dependencies of MTS. Informer (Zhou et al., 2021) develops a probsparse self-attention mechanism for long sequence forecasting. Autoformer (Wu et al., 2021) proposes a decomposition architecture with Auto-Correlation mechanism to capture the long-range temporal dependency for forecasting. Pyraformer (Liu et al., 2021b) is designed to learn the multi-resolution representation of the time series by the pyramidal attention module to capture long-range temporal dependency for MTS forecasting. FEDformer (Zhou et al., 2022) designs a seasonal-trend decomposition with frequency-enhanced blocks to capture the long-range temporal dependency. Although Transformer based models have proved to be useful for MTS forecasting, a single Linear model is more effective (Zeng et al., 2023) in MTS forecasting tasks. Apart from the historical-value methods mentioned before, there is a MLP based time-index model DeepTime(Woo et al., 2023) achieving state of the art recently. Different from the historical-value methods, the time-index model takes as input time-index features such as datetime features to predict the value of the time series at that time step. Despite the input value is different, considering the state-of-the-art performance and the same output value, we add DeepTime as a baseline in our comparison. Since these models primarily focus on temporal dependencies, they usually ignore the various channel dependencies.

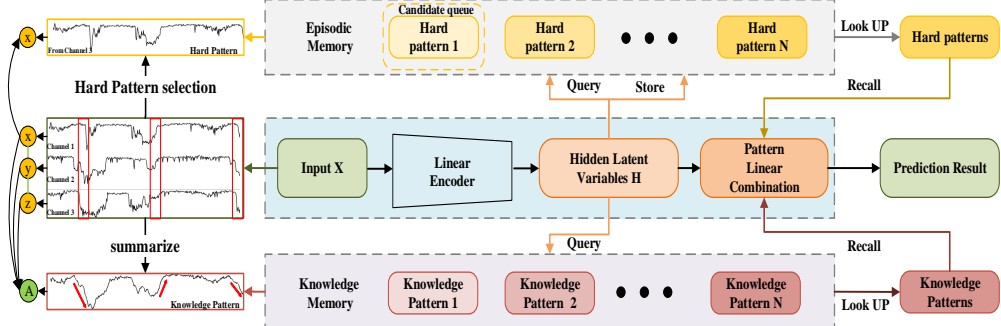

Figure 1: The framework of student-pattern memory net work.

## 2.2 CHANNEL DEPENDENCY FOR MTS FORECASTING

A multitude of models have been introduced to address evident channel dependency, employing various techniques such as preserving dimension information within latent embedding spaces and utilizing Convolutional Neural Networks (CNN) (Lai et al., 2018) or Graph Neural Networks (GNN) (Wu et al., 2019) to capture these dependencies. However, it's worth noting that many CNN and GNN-based models encounter challenges in effectively capturing long-range temporal dependencies. In response to this issue, Crossformer(Zhang & Yan, 2023) has recently emerged as a solution designed to overcome these limitations. Crossformer employs dimension-segment-wise embedding and a two-stage attention layer to capture temporal and channel dependency respectively. Nevertheless, certain challenges persist within the domain of MTS forecasting. For instance, some of the efficient modules mentioned face limitations when applied to Linear models which have been empirically proven to be more effective temporal models. Additionally, these modules may not fully capture the diverse array of channel dependencies present in the data. Therefore, distinct from the methods mentioned before, we propose a SPM-Net which can be considered as a general framework that can be applied to various deep learning models, utilizing Student-like memory to enhance their ability to capture channel dependency completely.

## 3 METHODOLOGY

### 3.1 PROBLEM DEFINITION

Defining the MTS as $\boldsymbol{x} = \{\boldsymbol{x}_1, \boldsymbol{x}_2, ..., \boldsymbol{x}_T\}$ , where $T$ is the duration of $\boldsymbol{x}$ and the observation at time $t$, $\boldsymbol{x}_t \in \mathbb{R}^N$, is a $N$ dimensional vector where $N$ denotes the number of channels, thus $\boldsymbol{x} \in \mathbb{R}^{T \times N}$. The aim of multivariate time series forecasting is to predict the future value of $\boldsymbol{x}_{T+1:T+T',n}$, where $T'$ is the number of time steps in the future.

### 3.2 THE HARD AND KNOWLEDGE PATTERNS FOR CAPTURING CHANNEL DEPENDENCIES

Within MTS, each channel usually represents a specific univariate time series which has its own temporal patterns, and at the same time different channels may also follow similar patterns, which are denoted as channel dependencies. For instance, in the context of weather MTS, channels like temperature and humidity often display correlated behavior, sharing analogous temporal trends. In the traditional way, they usually capture the evident channel dependency by using GCN 1 and CNN to mix the patterns between the two channels. However, to understand the relationship between those two channels completely, we think some reasons should be summarized and one of them may be the intrinsic variable, such as the rainfall in weather, which may not appear as one channel of MTS directly. Given this, we consider that there exists a latent channel dependency within MTS that is usually caused by the intrinsic variable, which has its own pattern that indicates the common features hidden behind different channels. To capture both the evident and latent dependencies, we introduce the hard and knowledge pattern information within MTS. Specifically, we call the selected representative patterns of each channel within MTS the hard pattern, and we propose a hard pattern selection strategy to choose them as shown in Fig 1. Different from the hard pattern, we name the pattern belonging to the intrinsic variables as the knowledge pattern, which should be summarized from the MTS as shown in Fig 1. We consider that mixing hard patterns of each channel is equal

to traditional methods which can capture the evident channel dependency and mixing knowledge patterns indicates finding the intrinsic variable which can capture the latent channel dependency.

To capture the channel dependency completely, the proposed SPM-Net uses a knowledge memory module to summarize the knowledge patterns of intrinsic variable and uses an episodic memory to select the hard patterns that appeared directly in the MTS. In order to mix the channel information provided by these patterns, we propose a recall strategy as shown in the second formula of 1 which realizes the message passing when different channels use the shared patterns to predict its own future value. Because each channel should not only focus on its own pattern but also consider the impact of other channels by using the shared patterns.

$$\boldsymbol{h}' = \boldsymbol{A}\boldsymbol{h}\boldsymbol{W}, \boldsymbol{h}' = \text{Recall}(\boldsymbol{h})\boldsymbol{W} \tag{1}$$

where the first formula means the traditional GCN method to capture the channel dependency; the second formula is our method, which uses a Recall 4 7 method to mix the channel information instead of the adjacent matrix $\boldsymbol{A} \in \mathrm{R}^{N \times N}$ of channels(N denotes the number of channels); $\boldsymbol{W} \in \mathrm{R}^{T \times d}$ is a matrix of a linear transformation(T is the input length and d is the output length); $\boldsymbol{h}' \in \mathrm{R}^{T \times d}$ represents the embedding mixed channel information.

### 3.3 STUDENT-PATTERN MEMORY NEURAL NETWORK

As we mentioned before, there are two ways to capture the channel dependency. One is mixing the temporal patterns of each channel within MTS directly, another is finding an effective way to summarize the temporal patterns of intrinsic variables between different channels. Given these two methods, we propose a student-pattern memory network, which is equipped with a knowledge memory module for summarizing the knowledge patterns hidden behind the MTS and an episodic memory module for capturing the hard pattern that appears as one channel of the MTS, for considering complete channel dependencies and achieving forecasting. In our SPM-Net, the final prediction results are computed by a Linear map function using the results obtained by recalling as shown in Fig 1.

$$\boldsymbol{h} = \text{Linear}(\boldsymbol{x}_{1:T,n}), \boldsymbol{x}_{T+1:T+T',n} = \text{Linear}\left(\text{Concat}(\boldsymbol{h}, \boldsymbol{M}^k, \boldsymbol{M}^e)\right) \tag{2}$$

where Linear means Linear map function, $\boldsymbol{M}^k, \boldsymbol{M}^e \in \mathrm{R}^{T \times N}$ indicates the results computed by formula 4 and 7; $\boldsymbol{h} \in \mathrm{R}^{T \times N}$ means the embedding of the currently entered MTS. The Concat means the concatenation between different tensors. The detail options of knowledge memory and episodic memory will be introduced in next section.

### 3.4 KNOWLEDGE-PATTERN MEMORY

As shown in the Fig 1, there are $N_1$ knowledge memory blocks in knowledge memory module denoted as $\boldsymbol{M}^k = \left\{\boldsymbol{M}_i^k\right\}_{i=1}^{N_1}$, where $i$ denotes the $i$th block and $k$ means the knowledge memory.

**Storage strategy:** The knowledge memory in the network can provide intrinsic pattern information for different channels to predict its own future value by summarizing some MTS samples. According to the fact that a common student usually knows nothing about the task at first, the initialization should be random and the memory should be learnable during training. To realize the communication of different channels, each knowledge memory block is designed to store the univariate temporal pattern named knowledge pattern, which can be recalled by any channel present in the currently entered MTS. Considering what we mentioned before, we initialize the n-th block (pattern) of knowledge memory with a learnable vector $\boldsymbol{M}_i^k$ sampled from a normal Gaussian distribution, which is a $T \times 1$-dimensional vector.

**Recall strategy:** To predict the future value of the currently entered MTS, the knowledge memory first calculate the attention score between each channel and each knowledge pattern stored in knowledge memory and then use an attention mechanism to aggregate these patterns into a specific knowledge-pattern memory $\boldsymbol{m}_i^k$ for predicting the future value, where the attention score can be calculated as:

$$\boldsymbol{h}_j^q = \text{Linear}(\boldsymbol{h}_j), \text{score}\left(\boldsymbol{M}_i^k, \boldsymbol{h}_j^q\right) = \text{Cosine}\left(\boldsymbol{M}_i^k, \boldsymbol{h}_j^q\right) \tag{3}$$

where score$(\cdot)$ means the attention score function defined by cosine similarity; $\boldsymbol{h}_j$ denotes the $j$th channel of MTS embedding $\boldsymbol{h}$; $\boldsymbol{h}_j^q$ represents a query vector of the $j$th channel of the currently entered MTS, which is used for calculating the attention score for combining the knowledge patterns.

Then, to get the specific knowledge-pattern memory for the currently entered MTS, we can further aggregate these knowledge patterns with their attention weights, formulated as:

$$\boldsymbol{m}_j^k = \sum_{i=1}^{N_1} \boldsymbol{\lambda}_{i,j} \boldsymbol{M}_i^k, \quad \boldsymbol{\lambda}_{i,j} = \frac{\text{score}(\boldsymbol{M}_i^k, \boldsymbol{h}_j)}{\sum_{i=1}^{N_1} \text{score}(\boldsymbol{M}_i^k, \boldsymbol{h}_j)} \tag{4}$$

where $\boldsymbol{m}_j^k \in \mathrm{R}^{T \times 1}$ has gathered the knowledge patterns information from the knowledge memory $\boldsymbol{M}$ for forecasting future MTS.

**Update strategy:** As we mentioned before, the specific knowledge-pattern memory is calculated by knowledge patterns stored in the knowledge memory module. When we predict the future value, we combine the specific knowledge-pattern memory $\boldsymbol{m}_i^k$ with the $\boldsymbol{h}_j$. In this manner, each learnable knowledge pattern within the knowledge memory takes into account the prediction results of each channel in the MTS during gradient backpropagation. This approach effectively achieves the goal of facilitating message passing between different channels. Further more, to keep memory items as compact as possible, at the same time as dissimilar as possible, we use two constraints (Gong et al., 2019) (Jiang et al., 2023), including a consistency loss L1 and a contrastive loss L2, denoted by

$$\begin{aligned} L_1 &= \sum_j^N \left\| \boldsymbol{h}_j^q - \boldsymbol{M}_j^{k,1} \right\|^2 \\ L_2 &= \sum_j^N \max\left\{ \left\| \boldsymbol{h}_j^q - \boldsymbol{M}_j^{k,1} \right\|^2 - \left\| \boldsymbol{h}_j^q - \boldsymbol{M}_j^{k,2} \right\|^2 + \lambda, 0 \right\} \end{aligned} \tag{5}$$

where $\boldsymbol{M}_j^{k,i} \in \mathrm{R}^{T \times 1}$ denotes the ith similar knowledge pattern; $\lambda$ denotes the margin between the first similar knowledge pattern and the second similar knowledge pattern; N indicates the number of MTS channels. More details about knowledge memory update strategy is described in Appendix B.1

## 3.5 EPISODIC-PATTERN MEMORY

As shown in the Fig 1, there are $N_2$ episodic memory blocks in episodic memory module denoted as $M^e = \{M_i^e\}_{N_2}$, where $i$ denotes the $i$th block and $e$ means the episodic memory.

**Storage strategy:** The design of episodic memory is inspired by previous works (Fortunato et al., 2019; Guo et al., 2020), and the main idea is to employ a memory module to collect a subset of representative data samples. In our model, we hope it can capture the evident channel dependency, thus we collect a subset of univariate series embeddings instead of storing the entire MTS embeddings. Different from knowledge memory, episodic memory tends to focus on storing more specific and representative univariate series within MTS directly like the student being compelled to remember the question previously struggled to understand. Specifically, the n-th block (pattern) of episodic memory is a vector of univariate series embedding selected from the past MTS embedding $\boldsymbol{h}^p$ named hard pattern which has been seen during training:

$$\boldsymbol{M}_i^e = \text{select}(\boldsymbol{h}^p) \tag{6}$$

where select$(\cdot)$ means the select function which is defined in update strategy; $\boldsymbol{h}^p$ denotes the embedding of past MTS. The episodic memory is empty at first because it has never seen any MTS.

**Recall strategy:** Similar to the recall method in the knowledge memory module, we first calculate the attention score according to the query vector. It is worth noting that the query vector in the episodic memory is one of the channels of currently entered MTS embedding $\boldsymbol{h}_j$, which is different from the query vector in the knowledge memory computed by the Linear map. Because the episodic memory directly stores the past channel embeddings, we can use the currently entered MTS embedding to be a query vector achieving the purpose of directly finding the related channel. Then we select top-k similar results for aggregating steps, which is a little different from the knowledge memory module. It is because the knowledge memory hopes each channel to make a difference to each memory block and the episodic memory module is designed to make each channel focus on part of other channels which have similar patterns to itself. The aggregation of different hard patterns can capture the evident channel dependency by mixing the pattern information from other channels. Considering that the hard patterns serve as references for forecasting, they should have a weighted score denoted as $\gamma$ to regulate the attention of our model. This weighting ensures that the model does not overly rely on specific patterns stored in memory, but rather emphasizes particular hard patterns that have occurred to a certain extent, akin to the discerning focus of a diligent student. The detail of recalling episodic memory is defined as follows:

$$\boldsymbol{m}_j^e = \gamma \sum_{i=1}^{K} \boldsymbol{\lambda}_{i,j} \boldsymbol{M}_i^e, \quad \boldsymbol{\lambda}_{i,j} = \frac{\text{score}(\boldsymbol{M}_i^e, \boldsymbol{h}_j)}{\sum_{i=1}^{K} \text{score}(\boldsymbol{M}_i^e, \boldsymbol{h}_j)} \tag{7}$$

the specific hard pattern memory $\boldsymbol{m}_j^e \in \mathrm{R}^{T \times 1}$ has gathered the relevant channel information through recalling the episodic memory $\boldsymbol{M}^e$.

**Update strategy:** Inspired by the fact that it is important for a student to remember those hard questions he/she often recalls, we design a hard pattern selection and frequency-based episodic memory update strategy. First about hard pattern selection, when a batch of MTS is coming during training, we compute the loss of each MTS in the batch and then we select the k MTS with the largest loss as hard examples. Typically, each hard example is separated into $n$ univariate series named hard patterns which are stored in $n$ episodic memory blocks $\{\boldsymbol{M}_i^e\}_n$ respectively, where $n$ means the number of channels in each MTS. This process defines the select function 6 mentioned before. The second point is about the frequency-based episodic memory update strategy. Undoubtedly, the most frequently accessed hard pattern holds the distinction of being the most representative pattern. In order to record how many times $\boldsymbol{M}_i^e$ was recalled during training, we use a frequency matrix denoted as $\boldsymbol{F}_{M_i^e}$. Then, for updating $\boldsymbol{M}_i^e$, if the current episodic memory block has not been filled yet, we will directly append the embedding of a new univariate series to the $\boldsymbol{M}_i^e$ as a new hard pattern. In a traditional way, we will update $\boldsymbol{M}_i^e$ by replacing the least used pattern in the block if its capability exceeds the limit $N_2$. Since the newly incoming patterns have a lower access frequency rate than previously incoming patterns, these new patterns are more likely to be replaced, and we define this issue as memory consolidation. To address this problem, we introduce a circular candidate queue within the episodic memory, specifically designed to store the hard patterns from the past $N_3$ occurrences, denoted as $Q^e = \{Q_i^e\}_{N_3}$, with the constraint $N_3 \leq N_2$. Patterns stored in memory are not replaced until this candidate queue overflows. Once the queue reaches its maximum capacity, a hard pattern with a low access frequency across the entire memory is replaced. This design grants new patterns more opportunities for access. It's worth noting that the frequency matrix $\boldsymbol{F}_{M_i^e}$ is reset to a zero matrix following each update of the episodic memory. The detailed memory updating process algorithm can be found in Appendix B.1.

### 3.6 THE COMBINATION OF TWO KINDS OF PATTERNS FOR MTS FORECASTING

After getting the embedding of the currently entered MTS $\boldsymbol{h}$ with the Linear map, we use the embedding $\boldsymbol{h}$ to look up the top-k-related hard patterns and knowledge patterns through computing the attention score. Then we can use the attention score to realize the recall strategy which can get the specific knowledge-pattern memory and specific hard pattern memory about the currently entered MTS. Because the specific knowledge-pattern memory capture the evident channel dependency and the specific hard pattern memory capture the latent channel dependency, we just need to fuse the two specific pattern memory in order to gain the complete channel dependency of the currently entered MTS. Finally, we use the mixed pattern memory and the embedding $\boldsymbol{h}$ to predict future value as shown in Fig 1. Therefore, the final task loss function can be formulated as :

$$Loss = \sum_{j=1}^{N} \left| \boldsymbol{X}_j - \boldsymbol{X}_j' \right|^2 + \alpha_1 L_1 + \alpha_2 L_2 \tag{8}$$

where the first part of $Loss$ denotes the MSE loss; $L_1, L_2$ is computed by 5; N indicates the number of MTS channels; $\boldsymbol{X}_j, \boldsymbol{X}_j'$ means the prediction and the ground truth of jth channel respectively; $\alpha_1, \alpha_2$ indicates the balance parameter of two constraints.

## 4 EXPERIMENTS

### 4.1 BASELINES AND EXPERIMENTAL SETTINGS

We evaluate the effectiveness of our model on eight datasets for MTS forecasting, including ETTh1, ETTh2, ETTm1, ETTm2, Illness, Weather, Electricity and Exhcange-rate (Zeng et al., 2023). The results are either quoted from the original papers or reproduced with the code provided by the authors. The way of data preprocessing is the same as (Zeng et al., 2023). We deploy two widely used metrics, Mean Absolute Error (MAE) and Mean Square Error (MSE) (Zhou et al., 2021) to measure the performance of MTS forecasting models. Six popular state-of-the-art historical-value methods are compared here, including: Crossformer (Zhang & Yan, 2023); Linear (Zeng et al., 2023); Fedformer (Zhou et al., 2022); Autoformer (Wu et al., 2021); Informer (Zhou et al., 2021) and LogTrans (Li et al., 2019). It is worth noting that DeepTime (Woo et al., 2023) the latest time-index model also achieves state-of-the-art on MTS forecasting tasks. To prove the MTS forecasting capability of our proposed historical-value model, we compare our model with those total seven methods. The summary statistics of these datasets, baselines and other implementation details are described in Appendix A.

## 4.2 MAIN RESULTS

Table 8 presents the overall prediction performance in which the best results are highlighted in boldface and second best results are underlined. Evaluation results demonstrate that our proposed method outperforms other state-of-the-art approaches in most settings and ranks top-2 in all settings. The detail analysis of error bar can be found in Appendix B.6.

Table 1: Multivariate long-term forecasting errors in terms of MSE and MAE, the lower the better. Multivariate results with predicted length as {24, 36, 48, 60} on the ILI dataset, the others as {96, 192, 336, 720}. Best results are highlighted in bold, and second best results are underlined.

| Methods | | Ours | | Linear | | DeepTime | | Crossformer | | Fedformer | | Autoformer | | Informer | | LogTrans | |
|---|---|---|---|---|---|---|---|---|---|---|---|---|---|---|---|---|---|
| Metrics | | MSE | MAE | MSE | MAE | MSE | MAE | MSE | MAE | MSE | MAE | MSE | MAE | MSE | MAE | MSE | MAE |
| ETTh1 | 96 | **0.370** | **0.392** | 0.375 | 0.397 | 0.372 | 0.398 | 0.420 | 0.440 | 0.376 | 0.419 | 0.449 | 0.459 | 0.865 | 0.713 | 0.878 | 0.740 |
| | 192 | **0.406** | **0.416** | 0.418 | 0.429 | 0.408 | 0.420 | 0.532 | 0.514 | 0.420 | 0.448 | 0.500 | 0.482 | 1.008 | 0.792 | 1.037 | 0.824 |
| | 336 | **0.438** | **0.446** | 0.479 | 0.476 | 0.443 | 0.449 | 0.440 | 0.461 | 0.459 | 0.465 | 0.521 | 0.496 | 1.107 | 0.809 | 1.238 | 0.932 |
| | 720 | **0.470** | **0.489** | 0.624 | 0.592 | 0.485 | 0.497 | 0.519 | 0.524 | 0.506 | 0.507 | 0.514 | 0.512 | 1.181 | 0.865 | 1.135 | 0.852 |
| ETTh2 | 96 | **0.286** | **0.351** | 0.288 | 0.352 | 0.290 | 0.353 | 1.140 | 0.772 | 0.346 | 0.388 | 0.358 | 0.397 | 3.755 | 1.525 | 2.116 | 1.197 |
| | 192 | **0.372** | **0.410** | 0.377 | 0.413 | 0.390 | 0.420 | 1.784 | 1.021 | 0.429 | 0.439 | 0.456 | 0.452 | 5.602 | 1.931 | 4.315 | 1.635 |
| | 336 | **0.429** | **0.454** | 0.452 | 0.461 | 0.489 | 0.486 | 2.640 | 1.400 | 0.496 | 0.487 | 0.482 | 0.486 | 4.721 | 1.835 | 1.124 | 1.604 |
| | 720 | 0.632 | 0.560 | 0.698 | 0.595 | 0.682 | 0.592 | 3.111 | 1.501 | **0.463** | **0.474** | 0.515 | 0.511 | 3.647 | 1.625 | 3.188 | 1.540 |
| ETTm1 | 96 | **0.299** | **0.343** | 0.308 | 0.352 | 0.307 | 0.351 | 0.320 | 0.373 | 0.379 | 0.419 | 0.505 | 0.475 | 0.672 | 0.571 | 0.600 | 0.546 |
| | 192 | **0.337** | **0.368** | 0.340 | 0.369 | 0.338 | 0.369 | 0.403 | 0.440 | 0.426 | 0.441 | 0.553 | 0.496 | 0.795 | 0.669 | 0.837 | 0.700 |
| | 336 | 0.372 | 0.389 | 0.376 | 0.393 | **0.366** | 0.391 | 0.551 | 0.525 | 0.445 | 0.459 | 0.621 | 0.537 | 1.212 | 0.871 | 1.124 | 0.832 |
| | 720 | 0.424 | **0.420** | 0.440 | 0.435 | 0.426 | 0.422 | 0.720 | 0.649 | 0.543 | 0.490 | 0.671 | 0.561 | 1.166 | 0.823 | 1.153 | 0.820 |
| ETTm2 | 96 | **0.165** | **0.257** | 0.168 | 0.262 | 0.166 | **0.257** | 0.250 | 0.347 | 0.203 | 0.287 | 0.255 | 0.339 | 0.365 | 0.453 | 0.768 | 0.642 |
| | 192 | **0.225** | **0.302** | 0.232 | 0.308 | **0.225** | **0.302** | 0.421 | 0.485 | 0.269 | 0.328 | 0.281 | 0.340 | 0.533 | 0.563 | 0.989 | 0.757 |
| | 336 | 0.290 | 0.350 | 0.320 | 0.373 | **0.277** | **0.336** | 1.276 | 0.805 | 0.325 | 0.366 | 0.339 | 0.372 | 1.363 | 0.887 | 1.334 | 0.872 |
| | 720 | **0.383** | **0.407** | 0.413 | 0.435 | **0.383** | 0.409 | 3.783 | 1.354 | 0.421 | 0.415 | 0.433 | 0.432 | 3.379 | 1.338 | 3.048 | 1.328 |
| Weather | 96 | **0.153** | **0.208** | 0.176 | 0.236 | 0.166 | 0.221 | 0.162 | 0.232 | 0.217 | 0.296 | 0.266 | 0.336 | 0.300 | 0.384 | 0.458 | 0.490 |
| | 192 | **0.197** | **0.252** | 0.218 | 0.276 | 0.207 | 0.261 | 0.207 | 0.277 | 0.276 | 0.336 | 0.307 | 0.367 | 0.598 | 0.544 | 0.658 | 0.589 |
| | 336 | **0.247** | **0.294** | 0.262 | 0.312 | 0.251 | 0.298 | 0.265 | 0.320 | 0.339 | 0.380 | 0.359 | 0.395 | 0.578 | 0.523 | 0.797 | 0.652 |
| | 720 | 0.318 | 0.346 | 0.326 | 0.365 | **0.301** | **0.338** | 0.388 | 0.391 | 0.403 | 0.428 | 0.419 | 0.428 | 1.059 | 0.741 | 0.869 | 0.675 |
| Electricity | 96 | **0.134** | **0.230** | 0.140 | 0.237 | 0.137 | 0.238 | 0.213 | 0.300 | 0.193 | 0.308 | 0.201 | 0.317 | 0.274 | 0.368 | 0.258 | 0.357 |
| | 192 | **0.150** | **0.247** | 0.153 | 0.250 | 0.152 | 0.252 | 0.290 | 0.351 | 0.201 | 0.315 | 0.222 | 0.334 | 0.296 | 0.386 | 0.266 | 0.368 |
| | 336 | **0.166** | **0.264** | 0.169 | 0.268 | 0.166 | 0.268 | 0.348 | 0.389 | 0.214 | 0.329 | 0.231 | 0.338 | 0.300 | 0.394 | 0.280 | 0.380 |
| | 720 | **0.201** | **0.297** | 0.203 | 0.301 | 0.201 | 0.302 | 0.404 | 0.423 | 0.246 | 0.355 | 0.254 | 0.361 | 0.373 | 0.439 | 0.283 | 0.376 |
| Exchange | 96 | **0.081** | **0.205** | 0.082 | 0.207 | **0.081** | **0.205** | 0.256 | 0.367 | 0.148 | 0.278 | 0.197 | 0.323 | 0.847 | 0.752 | 0.968 | 0.812 |
| | 192 | 0.159 | 0.295 | 0.167 | 0.304 | **0.151** | **0.284** | 0.469 | 0.508 | 0.271 | 0.380 | 0.300 | 0.369 | 1.204 | 0.895 | 1.040 | 0.851 |
| | 336 | **0.299** | 0.416 | 0.328 | 0.432 | 0.314 | **0.412** | 0.901 | 0.741 | 0.460 | 0.500 | 0.509 | 0.524 | 1.672 | 1.036 | 1.659 | 1.081 |
| | 720 | **0.786** | 0.675 | 0.964 | 0.750 | 0.856 | **0.663** | 1.398 | 0.965 | 1.195 | 0.841 | 1.447 | 0.941 | 2.478 | 1.310 | 1.941 | 1.127 |
| Iliness | 24 | **1.890** | **0.960** | 1.947 | 0.985 | 2.425 | 1.086 | 3.110 | 1.179 | 3.228 | 1.260 | 3.483 | 1.287 | 5.764 | 1.677 | 4.480 | 1.444 |
| | 36 | **2.086** | **1.007** | 2.182 | 1.036 | 2.231 | 1.008 | 3.429 | 1.222 | 2.679 | 1.080 | 3.103 | 1.148 | 4.755 | 1.467 | 4.799 | 1.467 |
| | 48 | **1.840** | **0.976** | 2.256 | 1.060 | 2.230 | 1.016 | 3.451 | 1.203 | 2.622 | 1.078 | 2.669 | 1.085 | 4.763 | 1.469 | 4.800 | 1.468 |
| | 60 | **2.091** | 0.992 | 2.390 | 1.104 | 2.143 | **0.985** | 3.678 | 1.255 | 2.857 | 1.157 | 2.770 | 1.125 | 5.264 | 1.564 | 5.278 | 1.560 |

## 4.3 ABLATION STUDY

**Effect of Student-like Memory:** In our model, we have two key modules: the knowledge memory module and the episodic memory module. We conducted an ablation study on the ETTh1, ETTm2, and Weather datasets, as presented in Table 2. Here, "w/o" indicates the absence of a particular module, and "w/o both" refers to the Linear model without any memory modules, serving as the baseline. In the table, "ours" denotes the SPM-Net without any ablations. We analyze the results shown in Table 2. 1) It's evident that both the knowledge memory module and the episodic memory module contribute significantly to our model's performance, as indicated in rows 2 and 3 of Table 2. This demonstrates that both modules effectively capture channel dependencies, thereby enhancing the representation capacity of the Linear model. 2) Comparing the results on the ETTh1 and Weather datasets, it can be seen that the knowledge memory module exhibits greater universality than the episodic memory module, and this alignment with our design intention is expected. Specifically, the knowledge memory module is designed to capture correlations across all channels by summarizing intrinsic temporal patterns, while the episodic memory focuses on remembering channels that are challenging to predict. Consequently, the knowledge module can be more effective in datasets with a larger number of channels (e.g., the Weather dataset with 21 channels compared to the 7 channels on the ETTh1 and ETTm2 datasets). 3) Comparing row 1 (baseline) with rows 2 and 3, it's evident that the combined use of both memory modules improves model performance. This reaffirms that the two memory modules can indeed capture channel dependency from distinct perspectives: evident channel dependency and latent channel dependency.

**Effect of Memory update strategy:** In our approach, we employ three key update strategies: hard pattern selection, pattern-based update, and frequency-based update. To assess the benefits of these structured memory update strategies, we conducted three ablation studies on the ETTh1, ETTm2, and Weather datasets. Here are the details of these experiments: 1)In the first experiment, the

Table 2: Ablation study on three datasets and the best results are highlighted in bold.

| Dataset | | ETTh1 | | | | ETTm2 | | | | Weather | | | |
|---|---|---|---|---|---|---|---|---|---|---|---|---|---|
| predict_length | | 96 | 192 | 336 | 720 | 96 | 192 | 336 | 720 | 96 | 192 | 336 | 720 |
| ours | MSE | **0.369** | **0.408** | **0.439** | **0.470** | **0.166** | **0.224** | **0.285** | **0.385** | **0.153** | **0.198** | **0.247** | **0.321** |
| | MAE | **0.391** | **0.417** | **0.448** | **0.492** | **0.259** | **0.305** | **0.349** | **0.408** | **0.209** | **0.253** | **0.296** | **0.350** |
| w/o knowledge | MSE | 0.371 | 0.410 | 0.445 | 0.474 | 0.168 | 0.229 | 0.299 | 0.395 | 0.174 | 0.215 | 0.260 | 0.323 |
| | MAE | 0.393 | 0.422 | 0.452 | 0.493 | 0.261 | 0.309 | 0.362 | 0.424 | 0.234 | 0.273 | 0.312 | 0.362 |
| w/o episodic | MSE | 0.372 | 0.411 | 0.442 | 0.499 | 0.168 | 0.230 | 0.307 | 0.385 | 0.155 | 0.199 | 0.250 | 0.322 |
| | MAE | 0.393 | 0.423 | 0.449 | 0.514 | 0.261 | 0.310 | 0.368 | 0.418 | 0.210 | 0.257 | 0.298 | 0.351 |
| w/o both | MSE | 0.381 | 0.422 | 0.477 | 0.512 | 0.173 | 0.231 | 0.321 | 0.415 | 0.175 | 0.217 | 0.262 | 0.325 |
| | MAE | 0.405 | 0.431 | 0.476 | 0.520 | 0.269 | 0.312 | 0.382 | 0.438 | 0.235 | 0.276 | 0.313 | 0.366 |
| w/o pattern-based | MSE | 0.377 | 0.420 | 0.451 | 0.489 | 0.173 | 0.235 | 0.313 | 0.418 | 0.175 | 0.216 | 0.260 | 0.323 |
| | MAE | 0.401 | 0.430 | 0.458 | 0.499 | 0.267 | 0.322 | 0.374 | 0.431 | 0.237 | 0.270 | 0.309 | 0.365 |
| w/o frequency-based and knowledge | MSE | 0.373 | 0.410 | 0.451 | 0.481 | 0.171 | 0.230 | 0.314 | 0.396 | 0.175 | 0.215 | 0.261 | 0.324 |
| | MAE | 0.394 | 0.425 | 0.455 | 0.490 | 0.267 | 0.311 | 0.368 | 0.427 | 0.236 | 0.273 | 0.311 | 0.362 |
| w/o hard pattern and knowledge | MSE | 0.371 | 0.411 | 0.451 | 0.481 | 0.171 | 0.232 | 0.300 | 0.396 | 0.175 | 0.215 | 0.261 | 0.324 |
| | MAE | 0.394 | 0.424 | 0.461 | 0.499 | 0.267 | 0.312 | 0.365 | 0.426 | 0.236 | 0.275 | 0.311 | 0.362 |

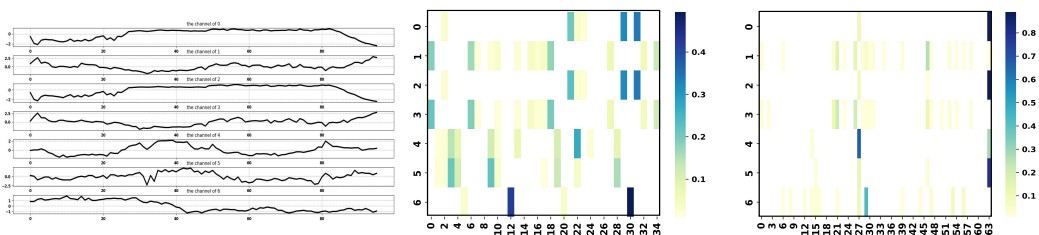

Figure 2: The visualization of input series(left), correlation matrix of knowledge memory(right) and episodic memory(middle) on ETTm1 dataset.

memory module stores a global MTS embedding vector $\boldsymbol{h} \in \mathbb{R}^{T \times n}$. This approach deviates from the pattern-based memory module proposed in this paper. The primary aim of this experiment is to assess the pattern-based memory module's capability to capture channel dependency. This study is labeled as "w/o pattern-based." 2).w/o Frequency-Based Episodic Memory Update: The second study focuses on the episodic memory update strategy and removes the frequency-based strategy in favor of a first-in, first-out (FIFO) queue-based memory update method. The goal is to assess the significance of the frequency-based update strategy, and this experiment is labeled as "w/o Frequency-based." 3).w/o Hard Pattern Selection: In the third study, we replace the hard pattern selection strategy with random choices. This investigation aims to highlight the importance of the hard pattern selection approach which is referred to as "w/o hard-pattern." The results of these experiments are presented in Table 2 and serve to confirm the effectiveness of the proposed memory update strategy. It's worth noting that the first study validates that our pattern-based memory module effectively captures channel dependency.

### 4.4 VISUALIZATION OF STUDENT-LIKE MEMORY

To further validate our SPM-Net's ability to capture channel dependencies, we conducted memory attention score visualization experiments on ETTm1 as shown in Fig 2. In this experiment, the number of channel is 7, the memory size of episodic memory is 35 and the memory size of knowledge memory is 64. These visualizations illustrate that similar channels recall related patterns to help them predict, which results in the similar prediction of different channels. This evidence confirms that our Student-like memory module is proficient at identifying shared patterns among different channels, thus effectively capturing various channel dependencies.

### 4.5 ADDITIONAL MODEL ANALYSIS

To demonstrate that our Student-like memory module can offer a more general and effective approach for capturing channel dependency, we conducted several additional experiments on different datasets. The results are presented in Figure 3, affirming that our Student-like memory module serves as a novel method for capturing channel dependency in MTS forecasting. Additional results for more datasets can be found in Appendix B.3 and B.4.

### 4.6 EFFECT OF HYPER-PARAMETERS

We evaluate the effect of three hyper-parameters: hard pattern memory weight $\gamma$, block number of knowledge memory and block number of episodic memory on the ETTh1 dataset.

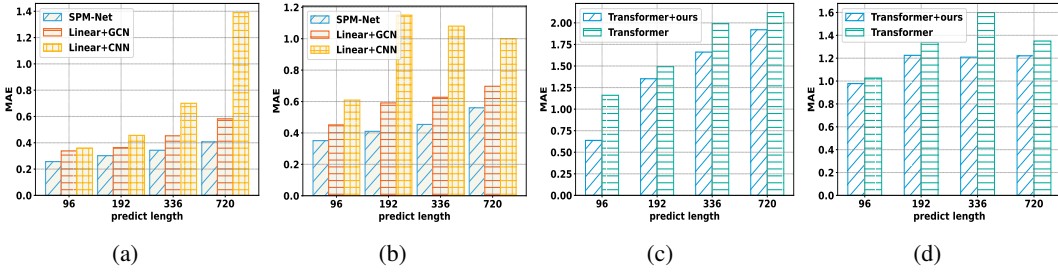

Figure 3: Additional model analysis on ETTm2 and ETTh2 dataset. (a)(b) Comparing our model with traditional methods used for capturing the channel dependency on ETTm2(left) and ETTh2(right). (c)(d) The generalization of our model with Transformer on ETTm2(left) and ETTh2(right).

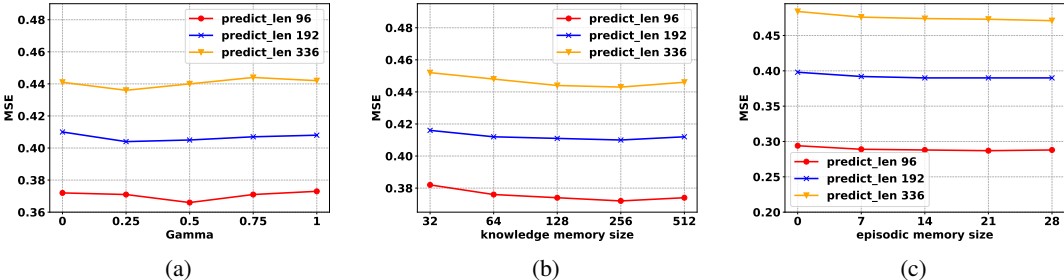

Figure 4: (a)-(b):The effect of gamma weight score and (a) and the size of knowledge memory (b) on ETTh1 dataset; (c) The effect of the size of episodic memory on ETTh2 dataset.

**Effect of hard example weight:** In Figure 4a, we varied the value of $\gamma$ from 0 to 1 and assessed the Mean Squared Error (MSE) across different prediction lengths on the ETTh1 dataset. As Fig 4a shows, the prediction result is the best when $\gamma$ is set as an appropriate value. When this value is too large, it means too much attention is paid to hard patterns that have appeared before. When this value is too small, it means that some extreme special cases that have appeared before are ignored.

**Effect of memory size on knowledge memory:** In Fig 4b, we set the memory size $N_1$ from 32 to 512 and evaluate MSE with different prediction lengths on ETTh1 dataset. To emphasize the impact of the memory size of the knowledge memory module, we conducted experiments without the episodic memory module. It becomes apparent that the knowledge memory size $N_1$ follows a non-linear relationship with performance. An excessively large memory size can be detrimental because the knowledge memory must learn to summarize patterns independently. In such cases, it might struggle to effectively capture useful patterns. Conversely, when the memory size is too small, it may not have the capacity to capture the essential patterns efficiently. Finding an optimal memory size is essential for striking the right balance between knowledge retention and model efficiency.

**Effect of memory size on episodic memory:** In Fig 4c, the memory size $N_2$ is set from 0 to 28 and evaluate MSE with different prediction lengths on ETTh2 dataset. In order to highlight the effect of the memory size of the episodic memory module, we remove the knowledge memory module. The results in Figure 4c demonstrate that as the memory size increases, the performance tends to improve. However, there is a point at which the performance gains start to diminish due to diminishing returns. This phenomenon underscores the effectiveness of the episodic memory module we designed. This module stores specific embeddings for challenging channels, and when the memory size is sufficiently large, it can record all these useful channel embeddings. It's important to note that as the memory size increases, the size of our candidate queue $N_3$ also needs to increase to prevent memory solidification, as mentioned earlier. Please see Appendix B.2 for experimental details of the candidate queue.

## 5 CONCLUSIONS

In this paper, we introduce a novel approach called SPM-Net for capturing channel dependency in multivariate time series forecasting tasks. SPM-Net comprises two crucial components: a knowledge memory module and an episodic memory module, both of which efficiently capture diverse channel dependencies. SPM-Net excels at extracting expressive representations from MTS, resulting in superior performance compared to other models in MTS forecasting tasks. Empirical results from various MTS forecasting tasks provide strong evidence of the effectiveness of our proposed model.

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

# A DETAILS OF EXPERIMENTS

## A.1 DATASETS STATISTICS

The detail of the datasets: (1) ETT dataset includes the time series of oil de-stationary factors and power load collected by electricity transformers from July 2016 to July 2018. ETTm1 and ETTm2 are recorded every 15 minutes, and ETTh1 and ETTh2 are recorded every 60 minutes. (2)Exchange dataset includes the the panel data of daily exchange rates from 8 countries from 1990 to 2016. (3) ILI dataset collects the ratio of influenza-like illness patients versus the total patients in one week, which is recorded weekly by Centers for Disease Control and Prevention of the United States from 2002 and 2021. (4) Weather dataset includes meteorological time series with 21 weather indicators collecteds from the Weather Station of the Max Planck Biogeochemistry Institute in 2020 every 10 minute. (5) Electricity dataset includes the electricity consumption of 321 customers recorded hourly from 2012 to 2014.

Table 3: Summary of statistics of datasets

| Datasets | Samples | channel number | Sample Rate |
|----------|---------|----------------|-------------|
| ETTh1 | 17420 | 7 | 60 min |
| ETTh2 | 17420 | 7 | 60 min |
| ETTm1 | 69680 | 7 | 15 min |
| ETTm2 | 69680 | 7 | 15 min |
| Exchange | 7588 | 8 | 1 day |
| Illness | 966 | 7 | 1 week |
| Weather | 52695 | 21 | 10 min |
| Electricity | 26304 | 321 | 60 min |

## A.2 EVALUATION METRICS

We use two evaluation metrics which is usually used in time series forecasting tasks to measure the performance of predictive models. Let $\boldsymbol{X}_{:,i} \in \mathbb{R}^{N \times 1}$ be the ground truth data of all channels at time step i, $\boldsymbol{X}'_{:,i} \in \mathbb{R}^{N \times 1}$ be the predicted values, and $\boldsymbol{\Omega}$ be indices of observed samples. The metrics are defined as follows.

Mean Absolute Error (MAE)

$$MAE = \frac{1}{|\Omega|} \sum_{i \in \Omega} \left| \boldsymbol{X}_{:,i} - \boldsymbol{X}'_{:,i} \right| \tag{9}$$

Mean Square Error (MSE)

$$MSE = \frac{1}{|\Omega|} \sum_{i \in \Omega} \left| \boldsymbol{X}_{:,i} - \boldsymbol{X}'_{:,i} \right|^2 \tag{10}$$

## A.3 BASELINE METHODS

The details of the baselines are as follows:

**Crossformer**: Crossformer is reproduced using the original paper's configuration in the official code.

**DeepTime**: DeepTime is reproduced using the original paper's configuration in the official code.

The results of other baselines are quoted from the paper of Linear model (Zeng et al., 2023).

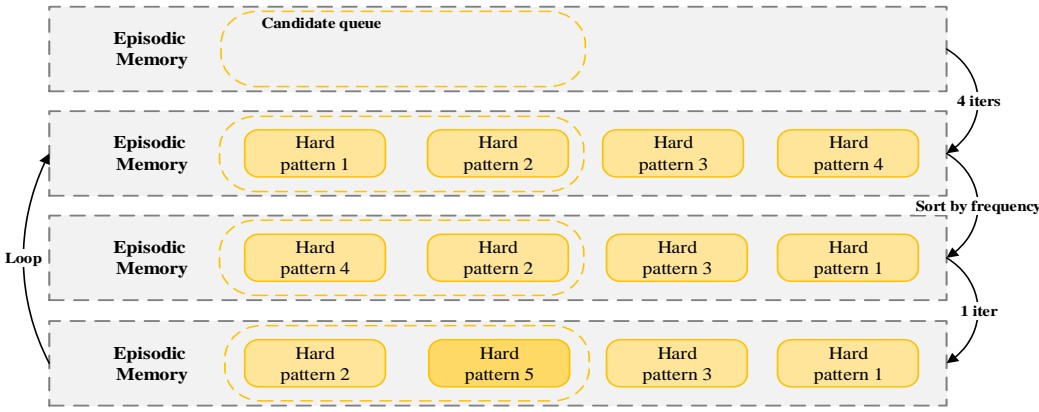

Figure 5: The details of episodic memory update strategy.

## A.4 IMPLEMENTATION DETAILS

**SPM-Net**: All the experiments are implemented with PyTorch and conducted on a single NVIDIA 3090 24GB GPU. Each model is trained by ADAM optimizer using MSE loss. For a fair comparison, the input length for the SPM-Net is consistent with Linear model (Zeng et al., 2023). For the illness dataset the input length is set to 104 and the input length of rest of datasets are set to 336.

## B ADDITIONAL MODEL ANALYSIS

### B.1 UPDATE STRATEGY ANALYSIS

#### B.1.1 EPISODIC MEMORY

In our approach, we make an initial assumption that a new hard pattern should be incorporated into the memory after each iteration. At the outset, the episodic memory is empty. As the memory fills up, we implement a sorting mechanism based on access frequency. Notably, only the first pattern in the candidate queue participates in this sorting process, as illustrated in Figure 5. When a new hard pattern arrives, it replaces the pattern occupying the first place in the candidate queue. Subsequently, the pattern in the first place exchanges positions with the pattern at the end of the candidate queue. This strategy effectively addresses the memory solidification issue that we previously discussed.

#### B.1.2 KNOWLEDGE MEMORY

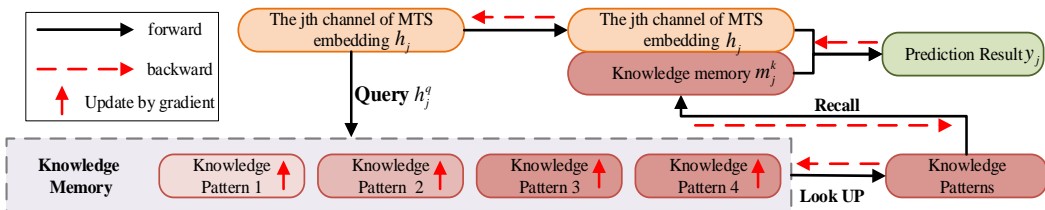

Figure 6: The details of knowledge memory update strategy.

The specific update process of the knowledge memory is elaborated in Figure 6. Since each knowledge pattern contributes to predicting the $j$th channel of the MTS $\boldsymbol{y}_j$, every pattern undergoes updates during the backward process and accumulates information from the $j$th channel of the MTS. Because

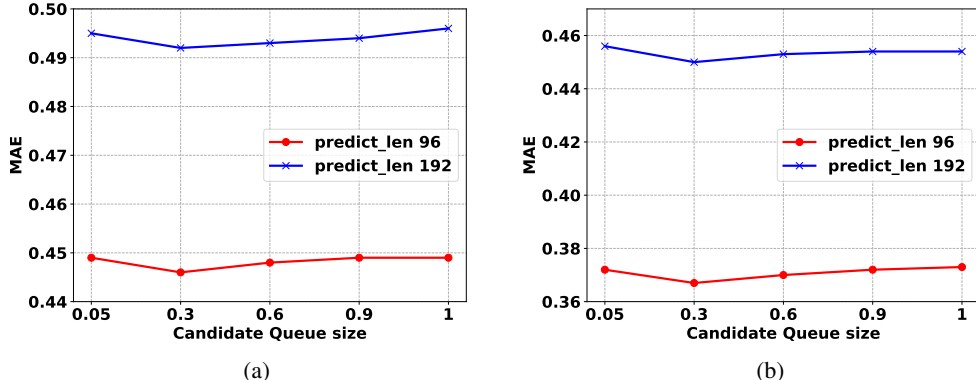

Figure 7: The effect of candidate queue size. (a) The result on the ETTh1 dataset. (b) The result on the ETTm2 dataset.

Table 4: Comparison with traditional methods on three datasets and the best results are highlighted in bold.

| Dataset | | ETTh1 | | | | Electricity | | | | Weather | | | |
|---|---|---|---|---|---|---|---|---|---|---|---|---|---|
| predict_length | | 96 | 192 | 336 | 720 | 96 | 192 | 336 | 720 | 96 | 192 | 336 | 720 |
| ours | MSE | **0.369** | **0.408** | **0.439** | **0.470** | **0.134** | **0.150** | **0.166** | **0.201** | **0.153** | **0.198** | **0.247** | 0.318 |
| | MAE | **0.391** | **0.417** | **0.448** | **0.492** | **0.230** | **0.247** | **0.264** | **0.297** | **0.209** | **0.253** | **0.296** | **0.346** |
| Linear+CNN | MSE | 0.447 | 0.542 | 0.877 | 0.912 | 0.364 | 0.443 | 0.400 | 0.341 | 0.158 | 0.207 | 0.250 | **0.305** |
| | MAE | 0.457 | 0.512 | 0.747 | 0.773 | 0.439 | 0.503 | 0.470 | 0.430 | 0.227 | 0.271 | 0.304 | 0.348 |
| Linear+GCN | MSE | 0.407 | 0.413 | 0.452 | 0.487 | 0.139 | 0.153 | 0.181 | 0.202 | 0.163 | 0.207 | 0.251 | 0.322 |
| | MAE | 0.423 | 0.422 | 0.455 | 0.495 | 0.238 | 0.253 | 0.287 | 0.300 | 0.238 | 0.275 | 0.313 | 0.364 |

all knowledge patterns are shared when utilized to forecast the outcomes of each channel, each pattern aggregates information from every channel, enabling it to capture latent channel dependencies effectively.

## B.2 THE EFFECT OF THE SIZE OF CANDIDATE QUEUE

When the candidate queue size accounts for 30% of episodic memory, the results as shown in the Fig 7 are the best on ETTh1 and ETTm2 dataset, which indicates that the design of candidate queue has a certain helpful effect to our model.

## B.3 COMPARISON WITH GCN AND CNN

To assess the efficiency of our model in capturing channel dependency, we introduced two straightforward baselines: a Linear model enhanced with CNN for channel dependency capture and a Linear model equipped with Adaptive GCN for channel dependency capture. Since MTS often lacks a graph structure to support GCN, we substituted the GCN module with Adaptive GCN, a method proven to be more efficient for MTS forecasting in RNN-based models (Bai et al., 2020) (Jiang et al., 2023). Using CNN for MTS embedding is a common approach to mixing channel information, as seen in previous works (Wu et al., 2021) (Zhou et al., 2022). Therefore, we adopted the same technique for the Linear+CNN model. The results presented in Table 4 confirm that our model is the superior and most effective approach for capturing channel dependency.

## B.4 THE GENERALIZATION OF SPM-NET

To prove our student-pattern memory Network can be a general framework, we design a new experiment, which combines our model with a Transformer model of which the decoder is replace with a Linear model. It is obvious that our student-like memory module can be used in Transformer model efficiently too as shown in the Table 5.

Table 5: The generalization experiment on three datasets and the best results are highlighted in bold.

| Dataset | | ETTh1 | | | | Electricity | | | | Weather | | | |
|---|---|---|---|---|---|---|---|---|---|---|---|---|---|
| predict_length | | 96 | 192 | 336 | 720 | 96 | 192 | 336 | 720 | 96 | 192 | 336 | 720 |
| Transformer+ours | MSE | **0.964** | **1.253** | **0.854** | **0.883** | **0.265** | **0.272** | **0.274** | **0.290** | **0.180** | **0.225** | **0.299** | **0.364** |
| | MAE | **0.810** | **0.920** | **0.704** | **0.757** | **0.363** | **0.371** | **0.373** | **0.385** | **0.271** | **0.305** | **0.367** | **0.403** |
| Transformer | MSE | 1.301 | 1.989 | 1.253 | 1.237 | 0.269 | 0.273 | 0.277 | 0.306 | 0.185 | 0.237 | 0.303 | 0.379 |
| | MAE | 0.931 | 1.176 | 0.919 | 0.861 | 0.365 | 0.373 | 0.376 | 0.397 | 0.272 | 0.319 | 0.369 | 0.418 |

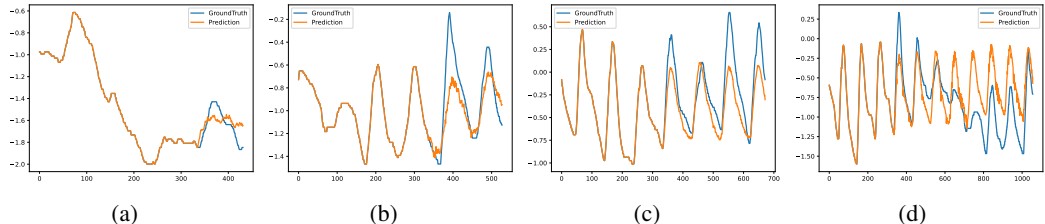

(a)      (b)      (c)      (d)

Figure 8: Prediction cases from the multivariate ETTm2 dataset of SPM-Net.

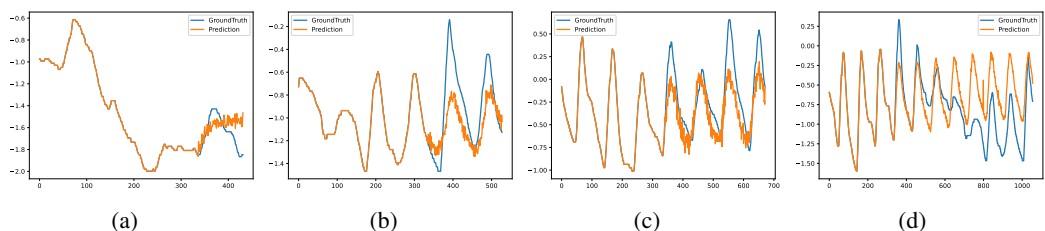

(a)      (b)      (c)      (d)

Figure 9: Prediction cases from the multivariate ETTm2 dataset of Linear model.

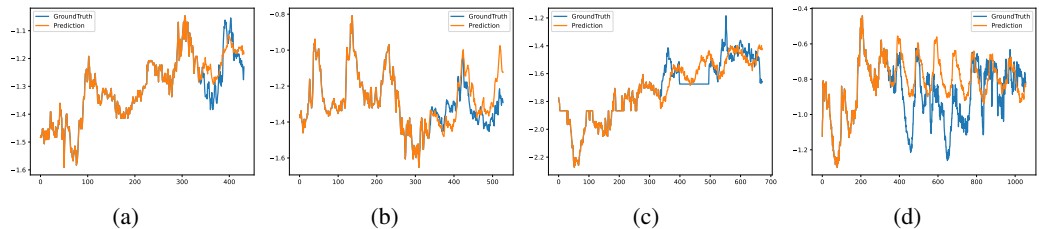

(a)      (b)      (c)      (d)

Figure 10: Prediction cases from the multivariate ETTm1 dataset of SPM-Net.

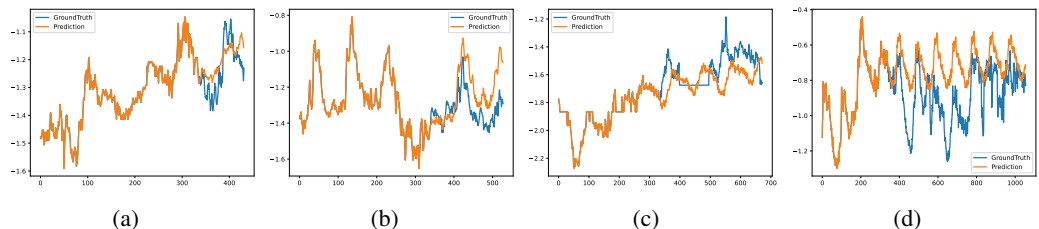

(a)      (b)      (c)      (d)

Figure 11: Prediction cases from the multivariate ETTm1 dataset of Linear model.

## B.5 PREDICTION VISUALIZATION

As shown in Figure 8 9 10 11, we plot the forecasting results from the test set of multivariate datasets ETTm1 and ETTm2 for comparison. Our model gives more accurate prediction than Linear model. Further more, SPM-Net can predict the change trends in time series better and smoother and closer to ground truth due to capturing the channel dependency.

Table 6: Error bar analysis on three datasets and the best results are highlighted in bold.

| Methods | | Ours | | Linear | | DeepTime | |
|---|---|---|---|---|---|---|---|
| Metric | | MSE | MAE | MSE | MAE | MSE | MAE |
| ETTh1 | 96 | **0.370±0.001** | **0.393±0.001** | 0.374±0.002 | 0.396±0.001 | 0.372±0.001 | 0.398±0.001 |
| | 192 | **0.405±0.002** | **0.419±0.002** | 0.420±0.001 | 0.429±0.001 | 0.406±0.002 | 0.420±0.001 |
| | 336 | **0.439±0.001** | **0.447±0.001** | 0.481±0.003 | 0.478±0.002 | 0.441±0.002 | 0.447±0.002 |
| | 720 | **0.471±0.001** | **0.490±0.001** | 0.511±0.001 | 0.520±0.001 | 0.485±0.003 | 0.499±0.002 |
| Weather | 96 | **0.154±0.001** | **0.208±0.001** | 0.176±0.001 | 0.238±0.002 | 0.166±0.001 | 0.223±0.001 |
| | 192 | **0.198±0.001** | **0.253±0.002** | 0.217±0.001 | 0.277±0.001 | 0.208±0.001 | 0.260±0.001 |
| | 336 | **0.248±0.001** | **0.295±0.002** | 0.268±0.007 | 0.321±0.008 | 0.251±0.002 | 0.297±0.002 |
| | 720 | 0.319±0.002 | 0.353±0.005 | 0.329±0.005 | 0.369±0.007 | **0.301±0.001** | **0.337±0.001** |
| Electricity | 96 | **0.135±0.001** | **0.232±0.002** | 0.141±0.001 | 0.239±0.001 | 0.137±0.001 | 0.238±0.001 |
| | 192 | **0.150±0.001** | **0.247±0.001** | 0.154±0.001 | 0.251±0.001 | 0.152±0.001 | 0.252±0.001 |
| | 336 | **0.166±0.001** | **0.265±0.001** | 0.169±0.001 | 0.267±0.001 | **0.166±0.001** | 0.268±0.001 |
| | 720 | **0.201±0.001** | **0.297±0.001** | 0.203±0.001 | 0.301±0.001 | **0.201±0.001** | 0.303±0.001 |

## B.6 MAIN RESULTS WITH STANDARD DEVIATIONS

To get more robust experimental results, we conduct main experiment three times with different random seeds on different datasets. For easier and fair comparison, the results reported in the main text are run with the fixed random seed 2021, which is the same as previous work. Table 6 shows the standard deviations.

## B.7 ADDITIONAL BASELINES

We add PatchTST, TimesNet and DLinear as new baselines as shown below. Considering our method, TimesNet and Dlinear do not use the 'revin normalization'(Kim et al., 2021), we use the result without the 'revin normalization' in PatchTST for fair comparison. Because 'revin normalization' is a key trick for any model to improve the performance. Prediction length is set to $24, 36, 48, 60$ for Iliness and $96, 192, 336, 720$ for others. The best results are highlighted in bold and the second best results are underlined. It can be clearly seen that our model can still surpass these models in most cases.

## B.8 ANALYSIS OF CHANNEL INDEPENDENCE

We combine our method with PatchTST and conduct tests on various datasets, we observe significant improvements as shown in Table. We add our memory module to the Linear decoder in PatchTST and maintain consistent experimental settings with it. The results of 'PatchTST without revin normalization' is replicated from their original paper(Nie et al., 2023) and the input length is set to 336. Therefore, we believe that the assumption of channel independence may not be entirely reasonable.

Table 7: Multivariate long-term forecasting errors in terms of MSE and MAE, the lower the better. Multivariate results with predicted length as {24, 36, 48, 60} on the ILI dataset, the others as {96, 192, 336, 720}. Best results are highlighted in bold.

| Methods | | Ours | | PatchTST | | TimesNet | | DLinear | |
|---|---|---|---|---|---|---|---|---|---|
| Metric | | MSE | MAE | MSE | MAE | MSE | MAE | MSE | MAE |
| ETTh1 | 96 | **0.370** | **0.392** | 0.388 | 0.412 | 0.384 | 0.402 | 0.375 | 0.399 |
| | 192 | 0.406 | **0.416** | 0.430 | 0.438 | 0.436 | 0.429 | **0.405** | **0.416** |
| | 336 | **0.438** | 0.446 | 0.454 | 0.458 | 0.491 | 0.469 | 0.439 | **0.443** |
| | 720 | **0.470** | **0.489** | 0.494 | 0.497 | 0.521 | 0.500 | 0.472 | 0.490 |
| ETTh2 | 96 | **0.286** | **0.351** | 0.313 | 0.374 | 0.340 | 0.374 | 0.289 | 0.353 |
| | 192 | **0.372** | **0.410** | 0.402 | 0.432 | 0.402 | 0.414 | 0.383 | 0.418 |
| | 336 | **0.429** | 0.454 | 0.448 | 0.465 | 0.452 | **0.452** | 0.448 | 0.465 |
| | 720 | 0.632 | 0.560 | 0.688 | 0.588 | **0.462** | **0.468** | 0.605 | 0.551 |
| ETTm1 | 96 | **0.299** | **0.343** | 0.308 | 0.358 | 0.338 | 0.375 | **0.299** | **0.343** |
| | 192 | 0.337 | 0.368 | 0.356 | 0.390 | 0.374 | 0.387 | **0.335** | **0.365** |
| | 336 | 0.372 | 0.389 | 0.389 | 0.411 | 0.410 | 0.411 | **0.369** | **0.386** |
| | 720 | **0.424** | **0.420** | 0.430 | 0.439 | 0.478 | 0.450 | 0.425 | 0.421 |
| ETTm2 | 96 | **0.165** | **0.257** | 0.167 | 0.257 | 0.187 | 0.267 | 0.167 | 0.260 |
| | 192 | 0.225 | **0.302** | 0.226 | 0.303 | 0.249 | 0.309 | **0.224** | 0.303 |
| | 336 | 0.290 | 0.350 | 0.301 | 0.348 | 0.321 | 0.351 | **0.281** | **0.342** |
| | 720 | **0.383** | **0.407** | 0.392 | 0.407 | 0.408 | 0.403 | 0.397 | 0.421 |
| Weather | 96 | **0.153** | **0.208** | 0.156 | 0.210 | 0.172 | 0.220 | 0.176 | 0.237 |
| | 192 | **0.197** | 0.252 | 0.199 | **0.250** | 0.219 | 0.261 | 0.220 | 0.277 |
| | 336 | **0.247** | **0.294** | 0.248 | **0.294** | 0.280 | 0.306 | 0.265 | 0.319 |
| | 720 | 0.318 | 0.346 | **0.313** | **0.342** | 0.365 | 0.359 | 0.323 | 0.362 |
| Electricity | 96 | 0.134 | 0.230 | **0.131** | **0.226** | 0.168 | 0.272 | 0.140 | 0.237 |
| | 192 | **0.150** | 0.247 | **0.150** | **0.244** | 0.184 | 0.289 | 0.153 | 0.249 |
| | 336 | **0.166** | **0.264** | 0.168 | 0.267 | 0.198 | 0.300 | 0.169 | 0.267 |
| | 720 | **0.201** | **0.297** | **0.201** | 0.298 | 0.220 | 0.320 | 0.203 | 0.301 |
| Exchange | 96 | **0.081** | 0.205 | 0.113 | 0.251 | 0.107 | 0.234 | **0.081** | **0.203** |
| | 192 | 0.159 | 0.295 | 0.410 | 0.498 | 0.226 | 0.344 | **0.157** | **0.293** |
| | 336 | **0.299** | 0.416 | 0.482 | 0.547 | 0.367 | 0.448 | 0.305 | **0.414** |
| | 720 | 0.786 | 0.675 | 0.987 | 0.756 | 0.964 | 0.764 | **0.643** | **0.601** |
| Iliness | 24 | **1.890** | **0.960** | 3.489 | 1.345 | 2.317 | 0.934 | 2.215 | 1.081 |
| | 36 | 2.086 | 1.007 | 4.629 | 1.550 | 1.972 | 0.920 | **1.963** | **0.963** |
| | 48 | **1.840** | 0.976 | 3.746 | 1.383 | 2.238 | **0.940** | 2.130 | 1.024 |
| | 60 | 2.091 | 0.992 | 5.174 | 1.622 | **2.027** | **0.928** | 2.368 | 1.096 |
| 1st count | | **37** | | 9 | | 6 | | 20 | |

Table 8: Multivariate long-term forecasting errors in terms of MSE and MAE, the lower the better. Multivariate results with predicted length as {24, 36, 48, 60} on the ILI dataset, the others as {96, 192, 336, 720}. Best results are highlighted in bold.

| Methods | | PatchTST+Ours | | PatchTST | |
|---|---|---|---|---|---|
| Metric | | MSE | MAE | MSE | MAE |
| ETTh1 | 96 | **0.384** | **0.408** | 0.388 | 0.412 |
| | 192 | **0.424** | **0.431** | 0.430 | 0.438 |
| | 336 | **0.445** | **0.448** | 0.454 | 0.458 |
| | 720 | **0.489** | **0.496** | 0.494 | 0.497 |
| ETTh2 | 96 | **0.301** | **0.357** | 0.313 | 0.374 |
| | 192 | **0.392** | **0.426** | 0.402 | 0.432 |
| | 336 | **0.386** | **0.430** | 0.448 | 0.465 |
| | 720 | **0.607** | **0.545** | 0.688 | 0.588 |
| ETTm1 | 96 | **0.303** | **0.355** | 0.308 | 0.358 |
| | 192 | **0.343** | **0.383** | 0.356 | 0.390 |
| | 336 | **0.378** | **0.405** | 0.389 | 0.411 |
| | 720 | 0.433 | **0.435** | **0.430** | 0.439 |
| ETTm2 | 96 | **0.167** | **0.256** | **0.167** | 0.257 |
| | 192 | **0.226** | **0.299** | **0.226** | 0.303 |
| | 336 | **0.285** | **0.341** | 0.301 | 0.348 |
| | 720 | **0.382** | **0.405** | 0.392 | 0.407 |
| Exchange | 96 | **0.101** | **0.240** | 0.113 | 0.251 |
| | 192 | **0.276** | **0.387** | 0.410 | 0.498 |
| | 336 | **0.443** | **0.501** | 0.482 | 0.547 |
| | 720 | **0.966** | **0.734** | 0.987 | 0.756 |
| Iliness | 24 | 2.876 | 1.223 | 3.489 | 1.345 |
| | 36 | 2.879 | 1.209 | 4.629 | 1.550 |
| | 48 | 2.985 | 1.238 | 3.746 | 1.383 |
| | 60 | 3.094 | 1.256 | 5.174 | 1.622 |

