# OpenReview forum: "Capturing The Channel Dependency Completely Via Knowledge-Episodic Memory For Time Series Forecasting"
_ICLR.cc/2024/Conference — Submitted to ICLR 2024_

### Official Review · Reviewer_a9EC · 2023-11-01

**Soundness:** 3 good
**Presentation:** 2 fair
**Contribution:** 2 fair
**Rating:** 3
**Confidence:** 5

**Summary:**

To model the channel dependency completely for MTS forecasting, this paper proposes a SPM-Net that uses a knowledge memory module to summarize the knowledge patterns of intrinsic variables and uses an episodic memory to store and select evident patterns in MTS. Instead of designing complicated models for long-term MTS forecasting, This paper formulates the problem as “prompt-enhanced” forecasting by treating encoded time series representation as queries and finding most similar hard and latent patterns. After concatenating the representations and recalled similar patterns as inputs, this paper uses a linear mapping function for prediction. Experiments on eight real-world datasets show the effectiveness of the model.

**Strengths:**

* The paper introduces a novel approach to capture channel dependencies in MTS forecasting, addressing both evident and latent dependencies.

* It is a very interesting work that formulates time series as exemplar (hard and latent) matching and simplifies the model architecture.

* The experiments and ablation studies are detailed to demonstrate the effectiveness of each module.

**Weaknesses:**

* Lack of the performance comparison of PatchTST and TimeNet, which are two SOTA baselines for LT-MTS forecasting from ICLR2023.

* No clear statements of the default values of \gamma_1 and \gamma_2. Although it has the effect of hard example weight in ablation study, we have two hyperparameters, which causes confusion.

* Have no concrete data preprocessing explanation, such as normalization, train/val/test splitting ratios for data.

* It is not convincing that channel dependencies are captured from the visualization of the model in 4.4.

**Questions:**

In the Recall strategy section, we use m to denote the aggregated knowledge patterns but never used again in other paper’s equations. Does it correspond to the output of Recall(M) ?  If yes, better to clear it in the equation 2.

Could you provide more detailed implementation details such as normalization protocols, splitting ratio, optimizer/scheduler etc.

---

> ### Author Response · Authors · 2023-11-16
> **To Reviewer a9EC**
>
> Thank you Reviewer a9EC for taking the time to read our paper and giving detailed feedback and questions. Please see below for our response to your specific questions, we hope they fully address any concerns and queries you have, and remain committed to address any further issues.
>
> **Q1: Lack of the performance comparison of PatchTST and TimeNet, which are two SOTA baselines for LT-MTS forecasting from ICLR2023.**
>
> A1: We would like to thank you for your comment and suggestion to compare with these two models. We provide the following table to compare the performance of the two models on multivariate setting, which is the same as 'response to all reviewers'. It can be seen that our model outperforms these models in most settings. The full efficiency comparison is provided in Table 7 of Appendix B.7.
>
> **Q2: No clear statements of the default values of $\gamma_1$ and $\gamma_2$. Although it has the effect of hard example weight in ablation study, we have two hyperparameters, which causes confusion.**
>
> A2: We have made the modifications in the article according to your suggestions and highlighted them in blue. $\gamma$ is the hard example weight and $\gamma_1, \gamma_2$ are replaced with $\alpha_1, \alpha_2$ which indicate the balance parameter of two constraints in Loss function.
>
> **Q3: It is not convincing that channel dependencies are captured from the visualization of the model in 4.4.**
>
> A3: We have made the modifications in the article and highlighted them in blue. The visualization is replaced with a correlation matrix. It is obvious that the similar channels recall the similar patterns which make similar channels have similar prediction.
>
> **Q4: In the Recall strategy section, we use m to denote the aggregated knowledge patterns but never used again in other paper’s equations. Does it correspond to the output of Recall(M) ? If yes, better to clear it in the equation 2.**
>
> A4: Thank you for your advice. It corresponds to the output of Recall(M), so we have corrected it as you pointed.
>
> **Q5: Could you provide more detailed implementation details such as normalization protocols, splitting ratio, optimizer/scheduler etc.**
>
> A5: We use Z-score Normalization to normalize the dataset. The splitting ration is set 7:1:2 for train, val and test set on Illness, weather, exchange and Electricity dataset. The splitting ration is set 3:1:2 for train, val and test set on ETTh1, ETTh2, ETTm1 and ETTm2 dataset. We use Adam as the optimizer. The scheduler is used the same as Linear model in their official code. All of these settings are the same as the Linear model and previous methods.

---

### Official Review · Reviewer_cd83 · 2023-11-02

**Soundness:** 2 fair
**Presentation:** 1 poor
**Contribution:** 2 fair
**Rating:** 3
**Confidence:** 3

**Summary:**

The paper proposes an approach for multivariate time series prediction. The approach is based on knowledge and episodic memory modules to capture channel dependencies across the time series. Authors propose strategies to populate and update each module based on the recall strategy. Linear model is then augmented with these memory modules for improved performance.

**Strengths:**

I found the memory approach interesting and potentially novel, although memory has been explored extensively in the context of RNNs. Authors also provide extensive empirical evaluation on multiple real world dataset and a detailed ablation study.

**Weaknesses:**

I found the paper very difficult to read due to grammar and references that point to pages rather than specific equations/figures, please consider revising. While the proposed approach is interesting I don't think the added complexity justifies the performance improvement over the linear model. From Table 1, the results for SPM-Net are nearly identical to Linear except for the long range prediction, and I suspect that most of them will not pass statistical significance so I don't think this method is ready for publication.

**Questions:**

In the ablation Table 2, why is performance worse than Linear when both memory modules are removed ("w/o both")? I thought that in this case the model would essentially be the same as Linear?

---

> ### Author Response · Authors · 2023-11-16
> **To Reviewer cd83**
>
> Thank you Reviewer cd83 for taking the time to read our paper and giving detailed feedback and questions. Please see below for our response to your specific questions, we hope they fully address any concerns and queries you have, and remain committed to address any further issues.
>
> **Q1: While the proposed approach is interesting I don't think the added complexity justifies the performance improvement over the linear model. From Table 1, the results for SPM-Net are nearly identical to Linear except for the long range prediction, and I suspect that most of them will not pass statistical significance so I don't think this method is ready for publication.**
>
> A1: In the paper of Linear model, they use a single Linear layer to predict the future value. However, in our method we use a Linear backbone which consists of a Linear encoder and a Linear decoder. Thus, the results for SPM-Net are nearly identical to Linear except for the long range prediction. We suggest using Table 2 to assess the effectiveness of our approach. We have done robustness checks in Appendix B.6 for your concerns.
> We also combine our method with PatchTST and conduct tests on various datasets to prove the effectiveness of our method in Appendix B.8 for your concerns.
> The results prove that our method is effective.
>
> **Q2: In the ablation Table 2, why is performance worse than Linear when both memory modules are removed ("w/o both")? I thought that in this case the model would essentially be the same as Linear?**
>
> A2: For fair comparison, the results used in Table 1 are replicated from the paper of the Linear model. However, in ablation study the 'without both' means the Linear backbone we used, which is equipped with a Linear encoder and a Linear decoder. In the paper of Linear model, they use a single Linear layer to predict the future value. Thus, the results in the Table 2 may be a little better or worse than the results in Table 1.

---

### Official Review · Reviewer_sqDM · 2023-11-04

**Soundness:** 2 fair
**Presentation:** 2 fair
**Contribution:** 2 fair
**Rating:** 5
**Confidence:** 3

**Summary:**

This paragraph discusses the importance and challenges of forecasting Multivariate Time Series (MTS), and introduces a memory-based forecasting method proposed in the research. The method aims to capture both latent and evident channel dependencies by utilizing knowledge and episodic memory modules. A pattern memory network is developed to effectively recall these memories and capture different channel dependencies comprehensively. Experimental results demonstrate the effectiveness of the proposed method.

**Strengths:**

1. The paper addresses the important problem of capturing channel dependencies in MTS forecasting, which is a crucial task in various domains such as weather prediction.

2. The proposed SPM-Net introduces two memory modules that provide a comprehensive approach to capturing both evident and latent channel dependencies.

3. The inclusion of recall strategies and attention mechanisms effectively mixes channel information from different patterns, enhancing the model's ability to capture dependencies.

4. The paper provides detailed explanations of the model architecture and the working principles of the memory modules, supported by equations and formulas.

5. The experimental results and analysis demonstrate the superior performance of the proposed SPM-Net compared to baselines, showcasing its effectiveness in capturing channel dependencies for MTS forecasting.

**Weaknesses:**

1. While the paper does a good job of introducing the model architecture and memory modules, more detailed explanations of certain components, such as the initialization of knowledge patterns and the selection process for hard patterns, could further enhance the reader's understanding.

2. The paper could benefit from more thorough discussions about the generalizability of the proposed SPM-Net across different types of MTS data and its limitations in handling noise or outliers.

3. It would be valuable to provide a comparative analysis of the computational complexity of the proposed approach compared to existing methods, as it could impact the practicality of the model.

**Questions:**

See the weakness part as above.

---

> ### Author Response · Authors · 2023-11-16
> **To Reviewer sqDM**
>
> Thank you Reviewer sqDM for taking the time to read our paper and giving detailed feedback and questions. Please see below for our response to your specific questions, we hope they fully address any concerns and queries you have, and remain committed to address any further issues.
>
> **Q1: While the paper does a good job of introducing the model architecture and memory modules, more detailed explanations of certain components, such as the initialization of knowledge patterns and the selection process for hard patterns, could further enhance the reader's understanding.**
>
> A1:  We introduce the initialization of knowledge patterns in the 'storage strategy' of section 3.4.  We typically select the top K channels with the highest prediction loss in each batch as hard  patterns.  For specific details on the update operations, please refer to the 'Update strategy' part in section 3.5. Due to the limited length of the main text, the selection process and update procsee for hard patterns are introduced in Appendix B.1.
>
> **Q2: The paper could benefit from more thorough discussions about the generalizability of the proposed SPM-Net across different types of MTS data and its limitations in handling noise or outliers.**
>
> A2: The generalizability of the proposed SPM-Net across different types of MTS data are shown in Table 4,5 in Appendix B.3,4.
> Due to the fact that our model does not specifically address noise and outliers, it does indeed have certain limitations. However, it's worth noting that this issue may also exist in other models, as most current models do not extensively analyze or handle noise and outliers. Our future work will be dedicated to designing models that are robust to noise and outliers, across different types of models.
>
> **Q3: It would be valuable to provide a comparative analysis of the computational complexity of the proposed approach compared to existing methods, as it could impact the practicality of the model.**
>
> A3: Thank you for your advice. We have added computational complexity of the proposed approach compared to existing methods as shown below:
> The original Transformer has $O(L^2)$ complexity on both time and space, where L is the number of input length. PatchTST use patch to reduce the L and achieve $O(N^2) (N<L)$ complexity on both time and space, where N is the number of input tokens. Dlinear is a Linear based model, so the complexity on both time and space is $O(L)$. The complexity of our model is caused by the calculation of attention score, which is $O(DM)$(D is the number of channels, M is the memory size). In most cases, D, M is much smaller than L, so the complexity of our model is smaller than the transformer based model.

---

### Official Review · Reviewer_D3M5 · 2023-11-04

**Soundness:** 2 fair
**Presentation:** 1 poor
**Contribution:** 2 fair
**Rating:** 5
**Confidence:** 5

**Summary:**

This work proposes a Student-like Pattern Memory Network (SPM-Net) for multivariate time series forecasting. The network introduces two memory modules to help describe channel dependencies in MTS. Following previous transformer works, experiments are performed on ETT, weather, electricity, exchange, and illness datasets.

**Strengths:**

- The use of episodic pattern memory from lifelong learning is interesting.
- The paper includes ablation studies on each component of SPM-Net.

**Weaknesses:**

Writing:
- The terminology used in the paper appears to be inappropriate, e.g., 'student-like pattern memory,' 'knowledge pattern memory,' and 'episodic pattern memory.'
- The word 'completely' in the title is inappropriate as there is a lack of evidence to demonstrate that the proposed model can **perfectly**  capture the complex dependencies. The proposed SPM-Net just introduces two memory modules to aid prediction.
- Symbols in all equations are not clearly introduced. For example, what are the sizes of W and A in (1)?
- All references are cited incorrectly. Most of them should be cited using \citep{}.
- There are numerous typos and grammar mistakes in the paper.

Model:
- Details of the combination part before outputting the final prediction results are missing.
- It would be beneficial to explain why memory can capture the dependencies and what advantages it has over graph structural learning methods."

Experiments:
- In your released source code, I noticed that in the test set dataloader, you set 'drop_last' to True (batch size=8). However, the Linear (Zeng et al. 2023) paper uses 'drop_last=False' and batch size=32. As you directly use their reported results of Linear for comparisons, there may be some inconsistencies in the experimental setups.
- The training objective (5) of the memory module (knowledge memory) is basically from the paper by Jiang et al. (2023). Thus, it is recommended to include this spatio-temporal baseline in the experiments. Additionally, it would be beneficial to include more commonly used spatio-temporal datasets in the experiments, such as METR-LA and PEMS-BAY, as suggested by Jiang et al. (2023).
> [Jiang et al. 2023] AAAI Spatio-Temporal Meta-Graph Learning for Traffic Forecasting
- Why choose Linear for comparison, not NLinear or DLinear (Zeng et al. 2023)?
- "Figure 2 is somewhat challenging to read. It would be better to display the correlation matrix found by the memories to demonstrate the channel dependencies.

Discussions on channel-independence
- One recent paper, PatchTST, utilizes channel-independence for multivariate time series. Could you provide some insights on the comparisons between the channel-independence and channel-dependency modeling methods?
> (PatchTST): A Time Series is Worth 64 Words: Long-term Forecasting with Transformers
- For some of the datasets in the paper where the units of variables differ, it is worth considering whether dependency modeling is necessary because PatchTST's performance seems to be good on them by channel independence.

**Questions:**

See Weaknesses.

---

> ### Author Response · Authors · 2023-11-16
> **To Reviewer D3M5**
>
> Thank you Reviewer D3M5 for taking the time to read our paper and giving detailed feedback and questions. Please see below for our response to your specific questions, we hope they fully address any concerns and queries you have, and remain committed to address any further issues.
>
> **About Writing:**
>
> **Q1: The terminology used in the paper appears to be inappropriate, e.g., 'student-like pattern memory,' 'knowledge pattern memory,' and 'episodic pattern memory.'**
>
> A1: We have made the modifications in the article according to your suggestions and highlighted them in blue.
>
> **Q2: The word 'completely' in the title is inappropriate as there is a lack of evidence to demonstrate that the proposed model can perfectly capture the complex dependencies. The proposed SPM-Net just introduces two memory modules to aid prediction.
> Symbols in all equations are not clearly introduced. For example, what are the sizes of W and A in (1)?**
>
> A2: About capturing the complex dependencies, we discuss it in 'About model' part Q2 and A2. About other questions, we have made the modifications in the article according to your suggestions and highlighted them in blue.
>
> **Q3: All references are cited incorrectly. Most of them should be cited using \citep{}.**
>
> A3: We have made the modifications in the article according to your suggestions.
>
> **Q4: There are numerous typos and grammar mistakes in the paper.**
>
> A4: We have try our best to correct the mistakes in the article according to your suggestions.
>
> **About Model**
>
> **Q1: Details of the combination part before outputting the final prediction results are missing.**
>
> A1: We have revised the combination and prediction in equation 2 mentioned in section 3.3. The 'Concat' means the concatenation between different tensors, which is usually implemented by torch.cat() in pytorch.
>
> **Q2: It would be beneficial to explain why memory can capture the dependencies and what advantages it has over graph structural learning methods."**
>
> A2:We believe that channel dependency arises from the fact that different channels can benefit from relevant information from other channels when predicting their own future results. Traditional CNN and GNN methods can achieve this operation, but our memory network captures channel dependency from different perspectives.
>
> Regarding knowledge memory: typically, different channels share knowledge memory, so the loss generated during prediction considers the influence of information from other channels, thus facilitating information transfer. Because the message passing between different channels is not direct, we name it latent channel dependency, which distinguishes it from graph learning models. The detail of message passing of knowledge memory can be found in Appendix B.1.2.
>
> Regarding episodic memory: episodic memory involves remembering representative channels, and each channel recalls several representative channels most similar to itself during prediction, enabling the intersection of information between the channel and representative channels. The key difference from graph learning models lies in the fact that our episodic memory aggregates information from the K nearest representative channels instead of aggregating all the information like GNN. This approach, to some extent, reduces unnecessary redundancy in the information.
>
> Our memory network achieves the capture of channel dependency by employing two different memory structures to aggregate effective information from different perspectives across channels. Through this approach, each channel considers the information from other channels when predicting its own results, thus capturing channel dependency.
>
> More details can be found in section 3.2 and the 'Effect of Memory update strategy' part of section 4.3. In ablation study, we introduce another data based memory based network to compare with our pattern based memory network, which proves our model can indeed capture channel dependency from distinct perspectives. Because in the data based memory network each channel has its own memory, which means different channels can not share the same pattern.

---

> ### Author Response · Authors · 2023-11-16
> **To Reviewer D3M5**
>
> **About Experiments:**
>
> **Q1: In your released source code, I noticed that in the test set dataloader, you set 'drop_last' to True (batch size=8). However, the Linear (Zeng et al. 2023) paper uses 'drop_last=False' and batch size=32. As you directly use their reported results of Linear for comparisons, there may be some inconsistencies in the experimental setups.
> The training objective (5) of the memory module (knowledge memory) is basically from the paper by Jiang et al. (2023). Thus, it is recommended to include this spatio-temporal baseline in the experiments. Additionally, it would be beneficial to include more commonly used spatio-temporal datasets in the experiments, such as METR-LA and PEMS-BAY, as suggested by Jiang et al. (2023).**
>
> A1: We set batch size=32 in the scripts when we train our model. The default setting in the code is not the final setting for our model. About 'drop_last problem': The results in the paper of Linear model are obtained from 'drop_last=True'. This issue can be found in their github issue(https://github.com/cure-lab/LTSF-Linear/issues/76). For fair comoparison, the previous methods and our model both use the 'drop_last=True' setting. Due to Jiang et al. (2023) focus on traffic forecasting, their model is based on RNN module which takes too much time to forecast long time series. Another reason is that the RNN based model is not good at long-term series forecasting. Considering that, we do not compare it with our model. About spatio time series: We have added a traffic dataset which records the road occupancy rates from different sensors on San Francisco freeways. The result is shown in 'response to all Reviewers'.
>
> **Q2: Why choose Linear for comparison, not NLinear or DLinear (Zeng et al. 2023)?**
>
> A2: Because we do not use 'revin normalization' or 'trend-seasonal decompose' operation which are proved to be effective to improve the performance of forecasting model, we think use Linear model is a more fair baseline. However, considering the issue you mentioned, we add DLinear model as a baseline in 'response to all reviewers'.
>
> **Q3: "Figure 2 is somewhat challenging to read. It would be better to display the correlation matrix found by the memories to demonstrate the channel dependencies.**
>
> A3: We have made the modifications in the article according to your suggestions and highlighted them in blue.
>
> **About Discussions on channel-independence**
>
> **Q1: One recent paper, PatchTST, utilizes channel-independence for multivariate time series. Could you provide some insights on the comparisons between the channel-independence and channel-dependency modeling methods?**
>
> A1: I think that this article makes sense to some extent. However, they have not provided compelling evidence to demonstrate that reintroducing channel dependency after channel independence through effective means will lead to performance degradation.
> By combining our method with PatchTST and conducting tests on various datasets, we observed significant improvements. Therefore, we believe that the assumption of channel independence may not be entirely reasonable.
> Our future work will be dedicated to address this issue completely.
>
> **Q2: For some of the datasets in the paper where the units of variables differ, it is worth considering whether dependency modeling is necessary because PatchTST's performance seems to be good on them by channel independence.**
>
> A2: We visualize different channels on different datasets and find that most of channels have similar trends as shown in section 4.4. Thus, we consider that the effective way to capture the channel dependency is necessary. Our viewpoint is to use channel independence to eliminate redundant correlation between channels and then effectively blend information between relevant channels using suitable methods, which may be more effective.
> By combining our method with PatchTST and conducting tests on various datasets, we observed significant improvements. Therefore, we believe that the assumption of channel independence may not be entirely reasonable.
> Our future work will be dedicated to address this issue completely.

---

### Author Response · Authors · 2023-11-16
**Response to all reviewers**

**Comment:**
We appreciate all the reviewers for their time to provide valuable comments and suggestions to improve our paper substantially.

**We find the positive feedback for our paper as follows:**

1.The paper introduces a novel approach to capture channel dependencies in MTS forecasting, addressing both evident and latent dependencies, which is a crucial task in various domains such as weather prediction.

2.The experiments and ablation studies are detailed to demonstrate the effectiveness of each module.

3.The inclusion of recall strategies and attention mechanisms effectively mixes channel information from different patterns, enhancing the model's ability to capture dependencies.

4.The paper provides detailed explanations of the model architecture and the working principles of the memory modules, supported by equations and formulas.

5.The experimental results and analysis demonstrate the superior performance of the proposed SPM-Net compared to baselines, showcasing its effectiveness in capturing channel dependencies for MTS forecasting.

**We make the following key updates to the paper to better address the reviewers’ concerns:**

1. We have made the modifications in the article according to your suggestions and highlighted them in blue.

2. The visualization in section 4.4 has been replaced with correlation matrix to better show the channel dependency between different channels.

3. We have added PatchTST, TimesNet and DLinear as new baselines as shown below. Considering our method, TimesNet and Dlinear do not use the 'revin normalization'(Kim et al.2022 ICLR), we use the result without the 'revin normalization' in PatchTST and directly replicate the results from their paper without making any artificial modifications. Because 'revin normalization' is a key trick for any model to improve the performance, we think use normalized PatchTST to compare with other models is not fair. Prediction length is set to ${{24,36,48,60}}$ for Iliness and ${{96, 192 , 336, 720}}$ for others. The best results are highlighted in bold and the second best results are underlined. The results in the table represent the average results for different prediction lengths. The full efficiency comparison is provided in Table 7 of Appendix B.7. It can be clearly seen that our model can still surpass these models in most cases.

4. Due to limited computational resources and time constraints, we combine our method with 'PatchTST withour revin' and conduct preliminary tests on various datasets, which shows significant improvements. Therefore, we believe that the assumption of channel independence may not be entirely reasonable. The experiment details can be found in Appendix B.8.

We hope that our replies and revisions address all reviewers' concerns, and we would appreciate any further comments or suggestions.

[Kim et al.2022] ICLR Reversible instance normalization for accurate time-series forecasting against distribution shift.

---

> ### Author Response · Authors · 2023-11-16
> **Response to all reviewers (main experiment results)**
>
> |   Methods   |     |    Ours   |           |  PatchTST |           |  TimesNet |           |  DLinear  |           |
> |:-----------:|:---:|:---------:|:---------:|:---------:|:---------:|:---------:|:---------:|:---------:|:---------:|
> |    Metric   |     |    MSE    |    MAE    |    MSE    |    MAE    |    MSE    |    MAE    |    MSE    |    MAE    |
> |    ETTh1    | Avg | **0.421** | **0.436** |   0.443   |   0.452   |   0.458   |   0.450   |   0.423   |   0.437   |
> |    ETTh2    | Avg |   0.430   |   0.444   |   0.461   |   0.464   | **0.414** | **0.427** |   0.431   |   0.447   |
> |    ETTm1    | Avg |   0.358   |   0.380   |   0.372   |   0.400   |   0.400   |   0.406   | **0.357** | **0.378** |
> |    ETTm2    | Avg | **0.266** |   0.329   |   0.267   | **0.326** |   0.291   |   0.333   |   0.267   |   0.334   |
> |   Illness   | Avg | **1.977** |   0.984   |   4.175   |   1.449   |   2.139   | **0.931** |   2.169   |   1.041   |
> |   Weather   | Avg | **0.229** | **0.275** |   0.230   |   0.276   |   0.259   |   0.287   |   0.246   |   0.300   |
> | Electricity | Avg | **0.163** |   0.260   | **0.163** | **0.259** |   0.192   |   0.295   |   0.166   |   0.263   |
> |   Exchange  | Avg |   0.331   |   0.398   |   0.498   |   0.513   |   0.416   |   0.443   | **0.297** | **0.378** |
> |   Traffic   | Avg | **0.431** | **0.295** |       0.451    |0.315           |   0.620   |   0.336   |   0.434   | **0.295** |
> |  1st  count |     |     9     |           |     3     |           |     3     |           |     5     |           |

---

> ### Author Response · Authors · 2023-11-16
> **Response to all reviewers (main experiment results)**
>
> |  Methods |     | PatchTST+Ours |           |  PatchTST |       |
> |:--------:|:---:|:-------------:|:---------:|:---------:|:-----:|
> |  Metric  |     |      MSE      |    MAE    |    MSE    |  MAE  |
> | Exchange |  96 |   **0.101**   | **0.240** |   0.113   | 0.251 |
> |          | 192 |   **0.276**   | **0.387** |   0.410   | 0.498 |
> |          | 336 |   **0.443**   | **0.501** |   0.482   | 0.547 |
> |          | 720 |   **0.966**   | **0.734** |   0.987   | 0.756 |
> |  Iliness |  24 |   **2.876**   | **1.223** |   3.489   | 1.345 |
> |          |  36 |   **2.879**   | **1.209** |   4.629   | 1.550 |
> |          |  48 |   **3.094**   | **1.256** |   3.746   | 1.383 |
> |          |  60 |   **2.985**   | **1.238** |   5.174   | 1.622 |
> |   ETTh1  |  96 |   **0.384**   | **0.408** |   0.388   | 0.412 |
> |          | 192 |   **0.424**   | **0.431** |   0.430   | 0.438 |
> |          | 336 |   **0.445**   | **0.448** |   0.454   | 0.458 |
> |          | 720 |   **0.489**   | **0.496** |   0.494   | 0.497 |
> |   ETTh2  |  96 |   **0.301**   | **0.357** |   0.313   | 0.374 |
> |          | 192 |   **0.392**   | **0.426** |   0.402   | 0.432 |
> |          | 336 |   **0.386**   | **0.430** |   0.448   | 0.465 |
> |          | 720 |   **0.607**   | **0.545** |   0.688   | 0.588 |
> |   ETTm1  |  96 |   **0.303**   | **0.355** |   0.308   | 0.358 |
> |          | 192 |   **0.343**   | **0.383** |   0.356   | 0.390 |
> |          | 336 |   **0.378**   | **0.405** |   0.389   | 0.411 |
> |          | 720 |     0.433     | **0.435** | **0.430** | 0.439 |
> |   ETTm2  |  96 |   **0.167**   | **0.256** | **0.167** | 0.257 |
> |          | 192 |   **0.226**   | **0.299** | **0.226** | 0.303 |
> |          | 336 |   **0.285**   | **0.341** |   0.301   | 0.348 |
> |          | 720 |   **0.382**   | **0.405** |   0.392   | 0.407 |

---

### Author Response · Authors · 2023-11-22
**Gentle reminder for further comments**

Dear Reviewers,

Hope everything is going well. As the discussion period is coming to an end very soon, we would like to send a gentle reminder to let you know that we look forward to hearing any further updates and thoughts you may have. Once again, thank you for your efforts and service.

---

### Meta-Review · Area_Chair_zon9 · 2023-12-08

**Metareview:**

The reviewers raised multiple concerns, including the experimental setting, comparison with other cross-channel methods and writing. Despite likely improvements from author feedback, a consensus on paper acceptance wasn't reached due to a brief discussion period and substantial revisions undertaken during this period.

**Justification For Why Not Higher Score:**

Despite likely improvements from author feedback, a consensus on paper acceptance wasn't reached due to a brief discussion period and substantial revisions undertaken during this period.

**Justification For Why Not Lower Score:**

n/a

---

### Decision · Program_Chairs · 2024-01-16

Reject